# A Field Theory-Based Novel Algorithm for Navigational Hazard Index

**Yihua Liu** * and **Yu Ma**

Merchant Marine College, Shanghai Maritime University, Shanghai 201306, China
* Correspondence: liuyh@shmtu.edu.cn

**Abstract:** Collision risk assessment is crucial for autonomous ships to identify risks and make decisions for collision avoidance. In this paper, a field theory-based navel algorithm is proposed to define and describe the risk of ship collision. Based on the field theory and the asymmetric Gaussian function, the ship collision risk field is constructed, and the field potential is used to characterize the navigational hazard index. The effectiveness and superiority of the algorithm are verified through study cases. The results show that the proposed algorithm can obtain the index continuously, reflect the risk level accurately, and provide reliable basis for the navigation of autonomous ships. Therefore, it has important application potential in the development of intelligence navigation.

**Keywords:** field theory; navigational hazard index; asymmetric Gaussian function

## 1. Introduction

With the increase in global commodity trade demand, water transport, as a more economical form of large cargo transportation, is also developing rapidly. At the same time, maritime traffic has become busier [1], leading to frequent maritime traffic accidents. According to the research on historical data of maritime traffic accidents [2], navigational casualties accounted for 54.4% of the casualties. In addition, it is generally believed that human factors account for 85–96% of all ship collision accidents [3,4]. Therefore, in order to improve the intelligent level of maritime transportation and weaken the influence of human factors, the research of autonomous ships has become a popular topic. In order to make the risk judgment of ship collision more objective and scientific, and eliminate subjectivity and randomness, it is of great significance to monitor the risk of ship collision globally.

Current research on the risk of ship collision by domestic and foreign scholars is mainly focused on the theory and practice of marine traffic engineering. Since automatic radar plotting aids (ARPAs) were applied to ship collision avoidance and navigation, ship pilots have used a distance-closest point of approach (DCPA) and a time-closest point of approach (TCPA) to assess the risk of ship collision, which is still in use and is the most direct method of determining collision risk. Kearon was the first to propose the need to calculate the risk of ship collision using the weighted methods of DCPA and TCPA [5]. H. Imazu et al. [6,7] put forward distance allowance and time allowance, analyzed the weight of these two elements on the impact of the risk of ship collision, and adopted the judgment formula in the form of Kearon formula to establish the collision risk assessment model [8]. However, such methods have low credibility as it does not take into account the inconsistent dimensionality of the input values. It is not comprehensive enough to judge the risk of ship collision only by considering two indicators, and the defined evaluation value of collision risk cannot reflect the risk of ships fully in practice. Then, by considering the randomness and fuzziness of the risk of ship collision, scholars use fuzzy mathematics to determine the collision risk between ships. Early scholars [9,10] only considered DCPA and TCPA, and generally adopted fuzzy inference to determine the

collision risk. Later scholars [11–15] began to realize the importance of multiple indicators and began to use fuzzy comprehensive evaluation methods to build a composite calculation model by comprehensively considering the ship speed ratio, the DCPA, the TCPA, the relative position, the distance between two ships, and other factors. Although the problem of dimension inconsistency has been solved, the problem of determining indicator weights has not been solved. With the proposal of artificial intelligence, some scholars have also considered the method of combining fuzzy theory with artificial neural network algorithms to calculate the risk of ship collision. This method trains the model through network learning and solves the problem preliminarily that makes is difficult to enumerate fuzzy rules one by one, so that the fuzzy system can be realized. Liu Kezhong and Wang Zhuxiang proposed to use a modified back propagation (BP) neural network algorithm to calculate the risk of ship collision [16]. Ji Yongqing took CPA and TCPA as the parameters to calculate the risk of ship collision, normalized the parameters, and calculated the collision risk using the method of establishing a neuro fuzzy system [17]. Although using the modified neural network to calculate the risk of ship collision can obtain good calculation accuracy, there are still some factors that affect the risk of ship collision not taken into account, which leads to problems such as lower accuracy in the established model and a longer modeling time. Moreover, some environmental factors, such as wind, flow, and visibility, have certain internal relations, so they are not independent of each other and are not suitable for such models. Afterwards, Li. B and Pang put forward an assessment method for the D-S method [18], but the calculation complexity of this method is relatively large. With the further development of artificial intelligence, the research of unmanned surface vehicles is also being paid more attention. Musa SJ, Lv Hongguang, and others have further studied the use of artificial intelligence methods to calculate the risk of ship collision [19,20]. In view of the above shortcomings, scholars thought of solving these problems by establishing targeted models for specific waters areas [21–24]. For some water areas with high collision accidents and high traffic density, some scholars have proposed a method to calculate the risk of ship collision for these water areas [25–28]. This method can better calculate the risk of ship collision, but it has several limitations. Because of this, most of them only study the navigation risk of port water areas.

Obviously, most of the above methods are still based on DCPA and TCPA. Szlapczynski and Szlapczynska [29] pointed out that in some cases, the methods based on DCPA and TCPA may not be sufficient to assess the risk of ship collision. Fujii and Tanaka [30] and Goodwin [31] first proposed an evaluation method based on the concept of ship domain (SD), which has become popular recently. From a microscopic perspective, a ship domain-based method is important for the risk analyses of ship collisions [32]. Zheng [33] proposed a SVM-based ship collision risk assessment algorithm by considering the quaternion ship domain (QSD) built by SD. Wang [34], based on the fuzzy neural network method, calculated the space collision risk. Liu [35] used the QSD to build a collision risk fuzzy evaluation model. Jiang [36] combined the QSD model with COLREGS to build a new collision risk model. Im and Luong [37] proposed a potential risk ship domain and calculated real-time ship collision risk. Zhang and Meng [38] proposed a probabilistic ship domain and assessed collision risks of different ships. It is worth mentioning that there are many applications of the collision avoidance algorithm based on artificial potential; for example, Lu [39] presented a real-time and deterministic path-planning method for autonomous ships, but there are few researches on collision risk.

As shown in Table 1, the symbol "$\sqrt{}$" demonstrates that the factor is taken into account when building the model; the symbol "$\times$" indicates that the factor is not considered. By comparative analysis, although the SD-based method can overcome the limitations of the traditional CPA-based method, the factors considered in some methods are not complete, which leads to a reduction in ship collision risk accuracy. Moreover, some methods do not explain the principle, which is not conducive to intuitive understanding. In addition, the SD-based method is not suitable for the collision risk identification of a multi-ship encounter situation. The research on collision risk under the situation of ship encounter

has become the inevitable trend of the development of intelligent ships. Therefore, the navigational hazard index (NHI) is proposed to define and describe the risk of ship collision in this paper. A field theory-based novel algorithm for navigational hazard index can solve the above problems, obtain the index continuously, reflect the risk level accurately, and provide a reliable basis for the navigation of autonomous ships.

**Table 1.** Some related factors taken into account in various models.

| Method by | Ship's Length | Ship's Speed | Two-Ship Encounter Situation | Multi-Ship Encounter Situation | Navigation Condition |
|---|---|---|---|---|---|
| Liu (2018) [35] | × | √ | × | × | × |
| Im and Lulong (2019) [37] | √ | √ | × | × | √ |
| Zhang and Meng (2019) [38] | √ | √ | × | × | × |
| Zheng (2020) [33] | √ | √ | √ | × | × |
| Jiang (2020) [36] | √ | √ | √ | × | × |

In this algorithm, the force on the ship is constructed by referring to the universal gravitation, and the asymmetric Gaussian function is introduced to establish the field. According to the principle of field superposition, the NHI between ships is calculated. Taking into account the International Regulation for the Preventing Collisions at Sea, the NHI is divided into four regions with different collision risk levels to indicate the risk level around the ship and show the ability to assess potential collision risks in real time. This algorithm is suitable for the dynamic environment, can provide support for collision avoidance decision making, and can deal with collision risk in the early stage.

The main contributions of this paper to ship collision risk research are as follows:

(1) Based on the force constructed on the ship, the idea of field theory and multiple factors, a ship collision field model is established. The proposed navigational hazard index (NHI), using the potential field characterizing the index, transforms the ship domain overlapping problem into the field superposition problem and calculates the value in real time. The proposed algorithm can display the dangerous water level of the ship collision dynamically.

(2) Considering the encounter situations, the NHI values are divided to make it reflect the risk level accurately. For the first time, the field-based concept is proposed to solve the problem of real-time evaluation of collision risk of ships in the process of encountering.

(3) By providing real-time decision support for unmanned ships in actual operation, an on-board anti-collision decision-making system is applied and the automation level of an autonomous ship is promoted.

The rest of the paper is organized as follows. Section 2 presents a field theory-based novel algorithm for navigational hazard index. Section 3 provides some instance verification. Discussions are carried out in Section 4. Finally, Section 5 gives some conclusions.

## 2. Materials and Methods

The concept of "field" was originally used to analyze the distribution and function of a certain physical quantity or mathematical function in space, and then gradually developed to analyze the distribution and function of other natural or social phenomena in space. There are both connections and differences between various fields. Fields can be expressed abstractly by mathematical models. Any object can form a field, and different objects produce different fields. The method based on field theory is widely used in the research of vehicle safety, but the research results in the safe navigation of ships are relatively few. Therefore, inspired by the field theory, this paper calculates the NHI of ships in different situations according to the interaction between ship safety fields.

### 2.1. Ship Area Division

When two ships look at each other, there are three types of two-ship encounter situations, namely head-on (①), crossing (②, ④), and overtaking (③), as shown in Figure 1.

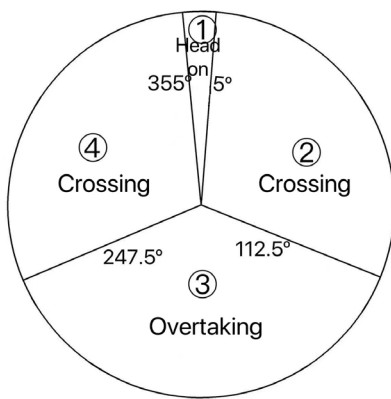

**Figure 1.** Classification of the ship encounter situation.

Since other ships take action when being overtaken, our ship's perspective does not consider the overtaking situation, as shown in Figure 2a (only the angle), and the area around the ship is divided into four areas accordingly, i.e., A, B, C, and D, as shown in Figure 2b.

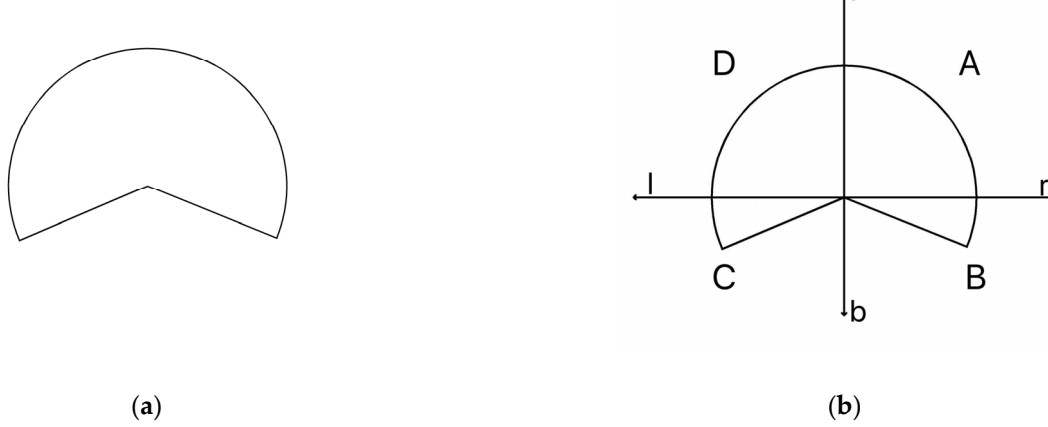

(**a**)                                                                 (**b**)

**Figure 2.** Regional division: (**a**) view angle of ship; (**b**) division of ship area.

### 2.2. The Force on the Ship

In the course of ship driving, the interaction between ship and ship, and ship and surrounding environment, which is the force described in this paper, including ship gravity, environmental force, and ship power.

#### 2.2.1. Ship Gravity

The ship gravity $F_c$ is constructed by referring to the universal gravitation and is used to describe the interaction between ships. The magnitude of this gravity is proportional to the product of their two masses and inversely proportional to the square of the distance between them. The formula is as follows:

$$F_c = c \frac{m_1 m_2}{r^2} \tag{1}$$

where $m_1$, $m_2$ are the mass of the ships, $r$ is the distance between two ships, and $c$ is the dangerous force constant. This paper takes $1 \text{ N·m}^2/\text{kg}^2$.

### 2.2.2. Environmental Force

The ship will be affected by the surrounding environment such as wind and flow during the driving process. The faster the speed is, the greater the force will be, and the bigger the risk of ship collision will be. This paper describes this effect with environmental force and refers to the momentum theorem $F\Delta t = m\Delta v$. The influence of wind is called wind force $F_w$, and the influence of flow is called flow force $F_f$. The formulas are as follows:

$$F_w = \frac{m_w \Delta v_w}{\Delta t_w} = \frac{\rho_w V_w \Delta v_w}{\Delta t_w} \tag{2}$$

$$F_f = \frac{m_f \Delta v_f}{\Delta t_f} = \frac{\rho_f V_f \Delta v_f}{\Delta t_f} \tag{3}$$

where $\rho_w$ is the density of the wind, and the wind is the flowing air, so the density of the air is the density of the wind, so $\rho_w = 1.29 \text{ kg/m}^3$. $V_w$ is the volume of wind, i.e., the volume of wind passing through the cross sectional area of the ship in $\Delta t$. $\rho_f$ is the density of the flow, i.e., the density of the sea water, so $\rho_f = 1025 \text{ kg/m}^3$. $V_f$ is the volume of flow, i.e., the volume of flow passing through the cross-sectional area of the ship in $\Delta t$. Simplification:

$$F_w = \frac{m_w \Delta v_w}{\Delta t_w} = \frac{\rho_w V_w \Delta v_w}{\Delta t_w} = \frac{\rho_w S_w \Delta t_w \Delta v_w \Delta v_w}{\Delta t_w} = \rho_w S_w \Delta v_w^2 \tag{4}$$

$$F_f = \frac{m_f \Delta v_f}{\Delta t_f} = \frac{\rho_f V_f \Delta v_f}{\Delta t_f} = \frac{\rho_f S_f \Delta t_f \Delta v_f \Delta v_f}{\Delta t_f} = \rho_f S_f \Delta v_f^2 \tag{5}$$

where $S_w$ is the cross-sectional area of the wind passing through the ship, and $S_f$ is the cross-sectional area of the flow passing through the ship.

### 2.2.3. Ship Power

In the course of ship driving, the ship itself has power, which we call ship power $F_s$. The formula is as follows:

$$F_s = \frac{m_s \Delta v_s}{\Delta t_s} \tag{6}$$

where $m_s$ is the ship's mass and $\Delta v_s$ is the change in the ship's own traveling speed in $\Delta t$.

### 2.3. Stress Analysis of Ship

The ship gravity and environmental force in each area of the ship are decomposed into four directions, namely $f$, $r$, $b$, and $l$, and are added together. The formulas are as follows:

$$F_f = F_A \sin\theta - F_D \cos\left(\theta - \frac{\pi}{2}\right) \tag{7}$$

$$F_l = F_D \sin\left(\theta - \frac{\pi}{2}\right) - F_C \cos(\theta - \pi) \tag{8}$$

$$F_b = -F_C \sin(\theta - \pi) - F_B \cos\left(\theta - \frac{3\pi}{2}\right) \tag{9}$$

$$F_r = -F_B \sin\left(\theta - \frac{3\pi}{2}\right) + F_A \cos\theta \tag{10}$$

where $F_f$, $F_l$, $F_b$, $F_r$ are the forces in the four directions of $f$, $l$, $b$, and $r$, respectively; $F_A$, $F_B$, $F_C$, $F_D$ are the forces in the four areas of $A$, $B$, $C$, and $D$, respectively; and $\theta$ is the angle of counterclockwise rotation from the $r$-axis.

### 2.4. Single-Ship Collision Risk Field Model

In this paper, the asymmetric Gaussian function is introduced to establish a single-ship collision risk field model. Assuming the ship position is $(x_0, y_0)$, the formulas of the ship collision risk field are described as follows:

$$\varphi(s) = \exp\left(-\left(\left(\frac{2(x-x_0)}{(1+\text{sgn}(x-x_0))\sigma_r+(1-\text{sgn}(x-x_0))\sigma_l}\right)^2 + \left(\frac{2(y-y_0)}{(1+\text{sgn}(y-y_0))\sigma_f+(1-\text{sgn}(y-y_0))\sigma_b}\right)^2\right)\right) \tag{11}$$

$$\sigma_i = F_i + F_s, \ i \in (f, l, b, r) \tag{12}$$

$$\text{sgn}(x - x_0) = \begin{cases} 1, & x - x_0 > 0 \\ 0, & x - x_0 = 0 \\ -1, & x - x_0 < 0 \end{cases} \tag{13}$$

$$\text{sgn}(y - y_0) = \begin{cases} 1, & y - y_0 > 0 \\ 0, & y - y_0 = 0 \\ -1, & y - y_0 < 0 \end{cases} \tag{14}$$

The spatial distribution of the ship collision risk field is shown in Figure 3a,b:

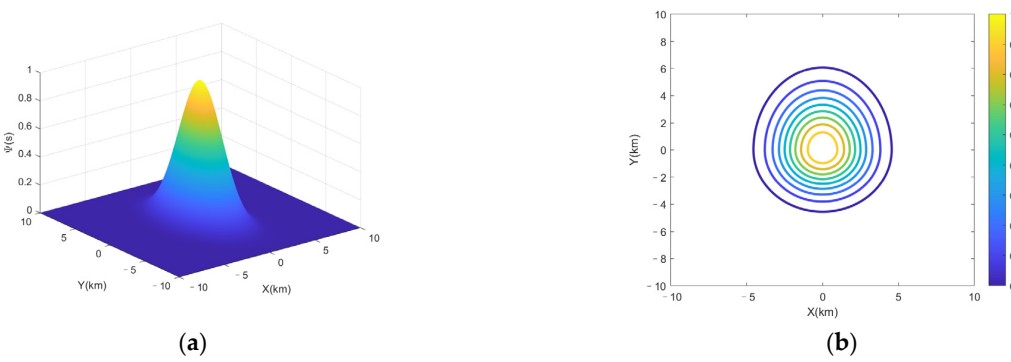

|   |   |
|---|---|
| (**a**) | (**b**) |

**Figure 3.** Schematic diagram of single-ship collision risk field: (**a**) three-dimensional view of single-ship collision risk field; (**b**) top view of single-ship collision risk field.

### 2.5. Navigational Hazard Index (NHI)

The NHI is the navigational hazard index between ships, and it ranges from 0 to 1. When the NHI is 0, no collision risk exists. Meanwhile, large values of the NHI mean high collision probability and vice versa.

In this paper, the field potential $\varphi(s)$ of the ship collision risk field is used to characterize the NHI, which is described as follows:

If there are no other ships in the ship collision risk field of this ship, i.e., if there is no overlap in the ship field, then:

$$\text{NHI} = 0 \tag{15}$$

If there are other ships in the ship collision risk field of this ship, i.e., if there is overlap in the ship field, then:

$$\text{NHI} = \frac{\sum_{k=1}^{n} \varphi(s_k)}{n} \tag{16}$$

where $\varphi(s_k)$ is the field potential of the $k_{th}$ ship in the ship collision risk field and n is the total number of other ships in the ship collision risk field.

### 2.6. Regionalization of the Collision Risk Index

According to the International Regulation for the Preventing Collisions at Sea, it is first necessary to judge the collision risk between ships, and then determine whether to

take collision avoidance action or change the current navigation conditions. All ships are free to take any action until the risk of collision exists.

According to the literature [40], the NHI is regionalized, as shown in Table 2.

**Table 2.** Regionalization of the ship collision risk index.

| Region | NHI | Definition Description |
|---|---|---|
| Safe Area | NHI = 0 | No risk of collision. |
| Caution Area | 0 < NHI ≤ 0.4 | The risk of collision is evident. |
| Action Area | 0.4 < NHI ≤ 0.8 | The risk of collision is obvious. |
| Emergency Braking Area | 0.8 < NHI ≤ 1 | The risk of collision is urgent. |

When the NHI is zero, the ship is located in the safe area. There is no collision risk and the ship is in the free driving stage. When the NHI is 0 < NHI ≤ 0.4, the ship is located in the caution area, and the collision risk begins to appear. The ship is advised to start paying attention to it and take substantive actions as soon as possible by judging the magnitude of the field potential of other ships in the collision risk field of the ship, so as to achieve a safety passing distance. When the NHI is 0.4 < NHI ≤ 0.8, the ship is located in the action area, and the collision risk is obvious. The ship is advised to take effective collision avoidance measures to avoid collision by judging the magnitude of the field potential of other ships in the collision risk field. When the NHI is 0.8 < NHI ≤ 1, the ship is located in the emergency braking area, and the collision risk is very urgent. The ship is required to take the action that is most helpful to avoid collision by judging the magnitude of the field potential of other ships in the collision risk field of the ship; otherwise, the collision will occur.

## 2.7. Diagram of Three Two-Ship Encounter Situation Types

When two ships look at each other, there are three two-ship encounter situation types. The ship collision risk field is shown in the figure below, in which Figure 4a–c show schematic diagrams of the ship risk field of the two ships in the head-on situation, the overtaking situation, and the crossing situation.

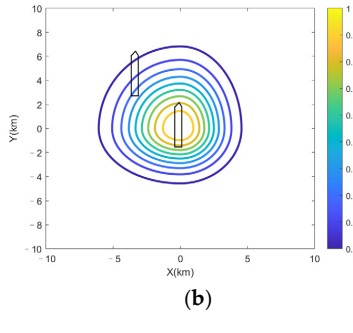

(a)

(b)

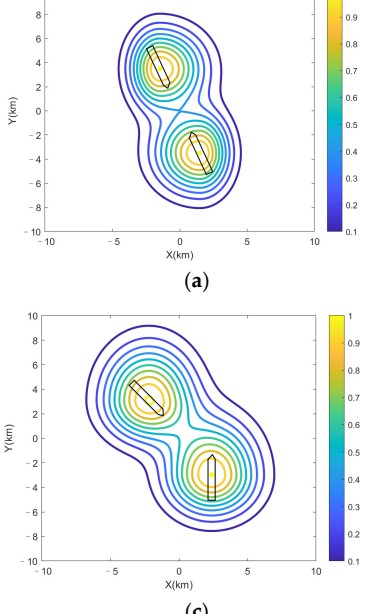

(c)

**Figure 4.** Schematic diagram of ship risk field in different situations: (**a**) schematic diagram of ship risk field in the head-on situation; (**b**) schematic diagram of ship risk field in the overtaking situation; (**c**) schematic diagram of ship risk field in the crossing situation.

### 3. Instance Verification

*3.1. Select Examples*

Yangshan Port is a large deep-water port located in the rugged islands of Shengsi County, Zhejiang Province, off the southeast coast of Shanghai, China. It is an important part of Shanghai Port, and the main part is located on the small Yangshan Mountain. With the economic development of Shanghai and the Yangtze River Delta, the container cargo sources generated by Shanghai and the Yangtze River Delta are increasing rapidly. The daily flow of ships is increasing, and the navigation density of ships is increasing, so the avoidance space between ships is greatly limited. Therefore, it is necessary to discuss the risk of ship collision in this water area.

Firstly, the AIS signals of ships near Yangshan Port are collected and the AIS signals are parsed, as partly shown in Figure 5.

| Time | MMSI | Lon(°) | Lat(°) | Course(°) | Speed(kn) | TrueHeading(°) |
|---|---|---|---|---|---|---|
| 2021/10/20 19:30 | 413234170 | 411.1695242 | 3310.958152 | 271.3 | 11.6 | 271 |
| 2021/10/20 19:36 | 413234170 | 408.8641013 | 3311.026737 | 270.5 | 13.3 | 270 |
| 2021/10/20 19:39 | 413234170 | 407.7468461 | 3311.058329 | 269 | 12.8 | 268 |
| 2021/10/20 19:40 | 413234170 | 407.147245 | 3311.061119 | 271.4 | 12.7 | 271 |
| 2021/10/20 19:46 | 413234170 | 405.1947506 | 3312.212294 | 314.5 | 12.5 | 314 |
| 2021/10/20 19:50 | 413234170 | 404.0275814 | 3313.302506 | 309.3 | 10.4 | 309 |
| 2021/10/20 19:54 | 413234170 | 403.1605968 | 3314.207361 | 316 | 10.6 | 315 |
| 2021/10/20 19:55 | 413234170 | 402.9224995 | 3314.458616 | 316.4 | 10.7 | 316 |
| 2021/10/20 20:00 | 413234170 | 401.9034157 | 3315.565691 | 314.3 | 11 | 314 |
| 2021/10/20 20:05 | 413234170 | 400.2532917 | 3316.593943 | 294.6 | 11.2 | 294 |
| 2021/10/20 20:11 | 413234170 | 398.5151325 | 3316.862437 | 261 | 10.6 | 260 |
| 2021/10/20 20:16 | 413234170 | 396.6969405 | 3316.749601 | 269.9 | 11.4 | 270 |
| 2021/10/20 20:17 | 413234170 | 396.3289624 | 3316.727923 | 264 | 11.6 | 264 |
| 2021/10/20 20:21 | 413234170 | 395.0317151 | 3316.442216 | 255 | 11.4 | 255 |
| 2021/10/20 20:26 | 413234170 | 393.2679081 | 3316.262557 | 261 | 12.1 | 260 |
| 2021/10/20 19:30 | 413360540 | 407.8057875 | 3311.181221 | 268.3 | 10.3 | 268 |
| 2021/10/20 19:36 | 413360540 | 406.1497477 | 3311.631393 | 301.8 | 11.4 | 301 |
| 2021/10/20 19:39 | 413360540 | 405.4718943 | 3312.189054 | 319 | 7.5 | 319 |
| 2021/10/20 19:40 | 413360540 | 405.2287034 | 3312.439007 | 310.4 | 8 | 311 |
| 2021/10/20 19:46 | 413360540 | 404.3005061 | 3313.352786 | 309.3 | 8.3 | 309 |
| 2021/10/20 19:50 | 413360540 | 403.3174356 | 3314.246071 | 313.6 | 8.7 | 313 |
| 2021/10/20 19:54 | 413360540 | 402.6203356 | 3314.867073 | 308 | 8.7 | 308 |
| 2021/10/20 19:55 | 413360540 | 402.403478 | 3315.025246 | 306 | 8.7 | 306 |
| 2021/10/20 20:00 | 413360540 | 401.2666023 | 3315.896072 | 303.3 | 9.5 | 303 |
| 2021/10/20 20:05 | 413360540 | 399.7725627 | 3316.462526 | 277.5 | 9.1 | 278 |

**Figure 5.** AIS data (partial).

AIS data contain a total of 1,048,576 data of all ships in Yangshan Port water areas in October 2021. The ship dynamic data are selected at a random time, and the ship MMSI, longitude and latitude, speed, course, and other data in the water area near Yangshan Port at that time are filtered from AIS data. Then, the data are further filtered, and the longitude and latitude data are converted into XY coordinates, so as to observe the positions of the ships and draw on the computer subsequently. The encounter situation between ships is judged by the position angle between ships on the map, the ships that form the two-ship encounter situation and the multi-ship encounter situation are screened out, and a group of ships is selected in the above situations randomly. According to the MMSI of the ship, the mass, speed, longitude and latitude, and other relevant factor values of the ship are queried, the dynamic data of the ship are filtered over one hour from AIS data, and the trajectory map of the ship is drawn. The change in the ship's NHI is then drawn in motion, and the case of whether the collision risk can be found earlier and more sensitively based on the actual situation is judged.

*3.2. Research on the Ship Collision Risk Index Based on Field Theory*

3.2.1. Examples of the Two-Ship Encounter Situation

At a time of 20:00:00, October 22, 2021, the ship MMSI, longitude and latitude, speed, course, and other data in the water area near Yangshan Port are filtered from AIS data. The

data of 90 ships are obtained through filtering, which indicates that 90 ships appear in the water area near Yangshan Port at that time. Then, further data filtering is carried out, and their longitude and latitude data are converted into XY coordinates, so as to observe the ship positions, as well as make subsequent judgments and calculations. The results are shown in Figure 6. Among them, the initial positions of "Ship a" (MMSI: 412424445) and "Ship b" (MMSI: 412442027) are (437.17km, 3318.14 km) and (436.95 km, 3318.18 km), respectively, and "Ship b" is located on the starboard side of "Ship a". It is judged preliminarily that the two ships are in the situation of crossing. The dynamic data of the ships are filtered in the final hour (from 19:30:00 on 22 October 2021 to 20:30:00 on 22 October 2021) from AIS data by their MMSI. Then, a trajectory map of the ship is drawn according to the filtered data. The trajectory map is shown in Figure 7a. It can be found that these two ships are in the crossing situation. According to the MMSI of the ship, the mass, speed, longitude and latitude, and other relevant factor values of the ship are queried, and the change in the NHI of the two ships is calculated during the situation, as shown in Figure 7b. (Note: The picture only shows the position and direction of the ship, not its actual size.)

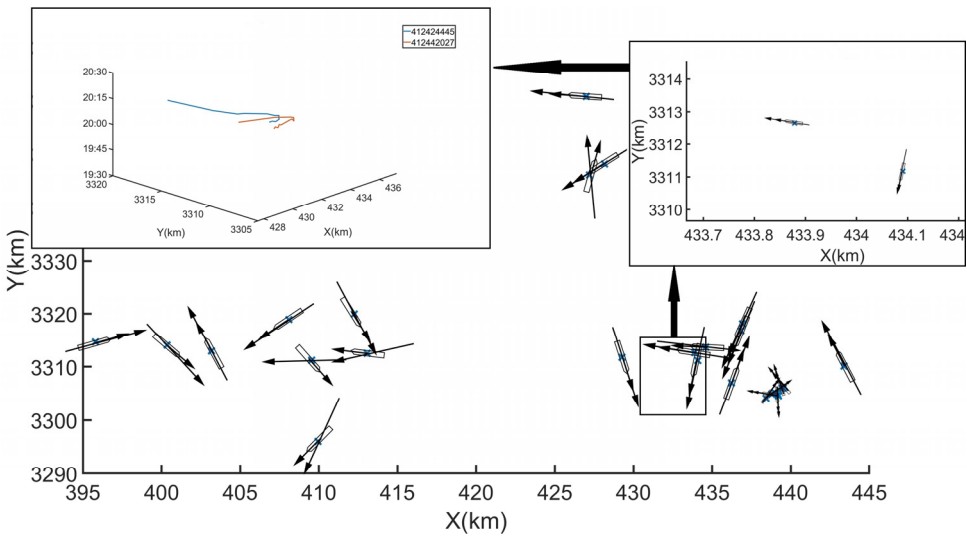

**Figure 6.** The crossing situation.

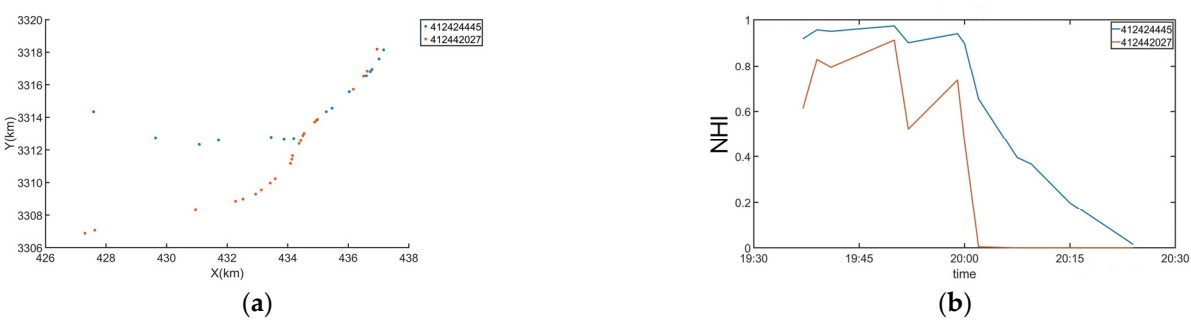

**Figure 7.** The crossing situation: (**a**) the trajectory of the crossing situation; (**b**) the NHI curve of the crossing situation.

As shown in Figure 7a, the two ships approach each other gradually and then separate gradually. In the process of the crossing situation, "ship b" approaches "ship a", gradually and the positions of the two ships change from Figure 8a to Figure 8b. "Ship a" and "ship b" mutually trigger the ship collision risk field, and "ship a" and "ship b" are located in each other's risk field. At this time, the NHIs of "ship a" and "ship b" in Figure 7b increased gradually, which is dangerous. The NHI reaches the peak value and changes from the action area to the emergency braking area. Subsequently, "ship a" and "ship b"

take measures one after another, with the speed slowing down gradually and the distance pulled away. The positions of the two ships are shown in Figure 8c, and the NHI of the two ships decreases gradually to the action area. Later, as the distance between the two ships decreases gradually, the NHI of the two ships starts to increase again, and there is a continuous collision risk during the crossing situation. After the crossing, the positions of the two ships are shown in Figure 8d. The distance between the two ships is pulled apart gradually, and the NHI shows a downward trend. "Ship a" is located in front of "Ship b" and accelerates. The positions of the two ships are shown in Figure 8e, so its NHI is reduced to a safe area. However, "Ship a" is still located in the ship collision risk field of "Ship b", so its NHI is reduced to a safe area more slowly. It can be seen that the algorithm proposed in this paper can identify the collision risk as early as possible, help to make anti-collision decisions, and improve navigation safety. This shows that if it is applied to the ship's autonomous navigation, when two ships are in the encounter situation, the ship operating system can obtain the real-time NHI through this model, evaluate the risk level around the ship, and make timely anti-collision actions to greatly avoid the occurrence of collision accidents.

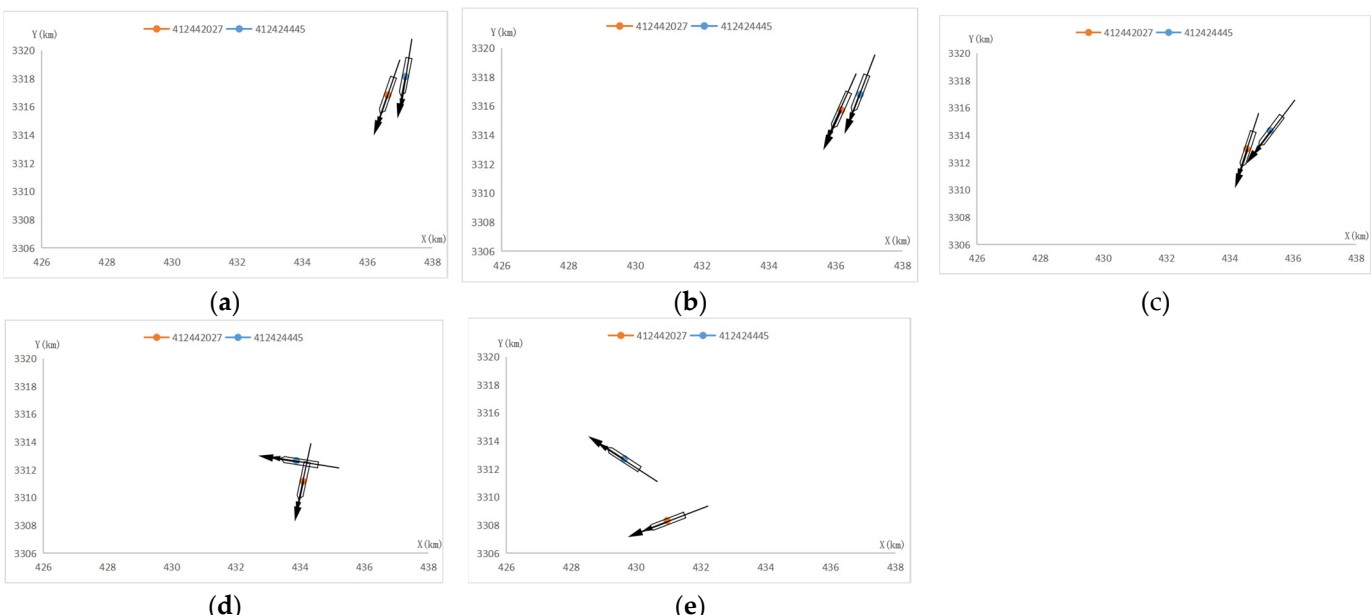

**Figure 8.** Position of two ships: (**a**) the two ships at 19:37; (**b**) the two ships at 19:41; (**c**) the two ships at 19:52; (**d**) the two ships at 20:00; (**e**) the two ships at 20:15.

### 3.2.2. Examples of Multi-Ship Encounter Situation

Another time of 20:00:00, 24 October 2021, is selected according to the above method, and the data of the ships are filtered in the water area near Yangshan Port from AIS data. Then, further data are filtered, and their longitude and latitude data are converted into XY coordinates, so as to observe the ship positions and make subsequent judgments and calculations. The results are shown in Figure 9. A group of ships is selected in a multi-ship encounter situation, and the dynamic data of the ships are filtered in the final hour (from 19:30:00 on 24 October 2021 to 20:30:00 on 24 October 2021) from the AIS data through their MMSI. Then, according to the filtered data, the trajectory map of the ship can be drawn. The trajectory map is shown in Figure 10a. Then, the change in the NHI of each ship in the multi-ship encounter situation is calculated, according to the values of the ship's mass, speed, and other relevant factors. The results are shown in Figure 10b.

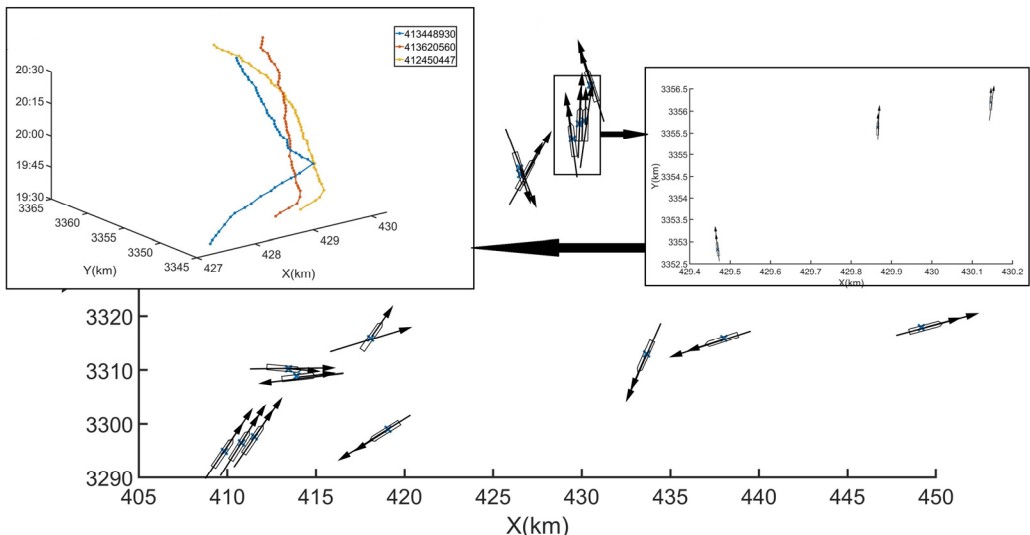

**Figure 9.** The multi-ship encounter situation.

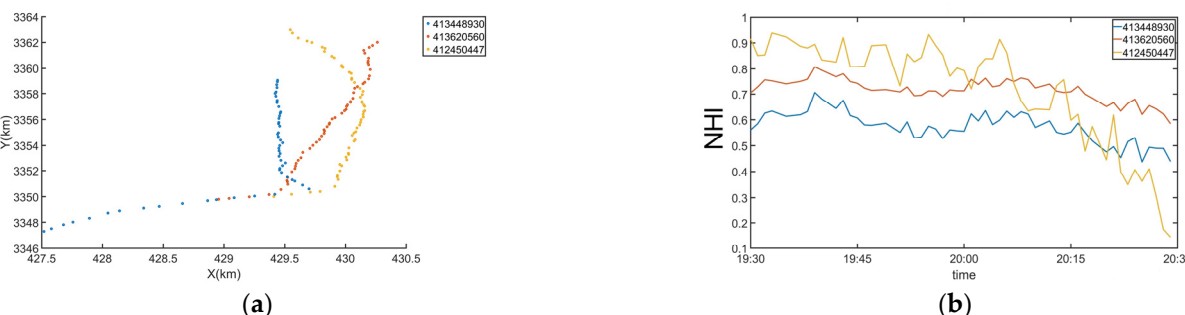

(**a**)                                                                                                   (**b**)

**Figure 10.** The multi-ship encounter situation: (**a**) the trajectory of the multi-ship encounter situation; (**b**) the NHI curve of the multi-ship encounter situation.

As shown in Figure 10a, the three ships first approach and then separate gradually. In this process, the three ships excite each other to create their ship collision risk fields. The three ships are close to each other, and the position changes from Figure 11a to Figure 11b. The NHI tends to rise. The ship represented by the yellow curve is dangerous, and the NHI reaches the emergency braking area. Then, the adjustment direction of the three ships is shown in Figure 11c, and the NHI of the three ships declines. Later, as shown in Figure 11d, the NHI of the three ships rose again as the ship spacing narrows gradually. There is a continuous collision risk during the encounter situation until it is completed. Then, as shown in Figure 11e, when the ship indicated by the yellow curve leaves, its NHI decreases to the safe area significantly, and the distance between the remaining two ships increases gradually, and their NHI shows a downward trend. It can be seen that the algorithm proposed in this paper can also calculate the NHI in real time even in the multi-ship encounter situation, assess and obtain the collision risk, help the pilot to make anti-collision decisions and operations in advance, leave time for the pilot to react and think, and improve the navigation safety greatly. Similarly, this shows that if it is applied to the autonomous navigation of ships, it is also applicable in the case of the multi-ship encounter situation, obtaining the NHI in real time, assessing the risk level around the ships, giving early warning in advance, and making timely anti-collision actions to avoid collision accidents.

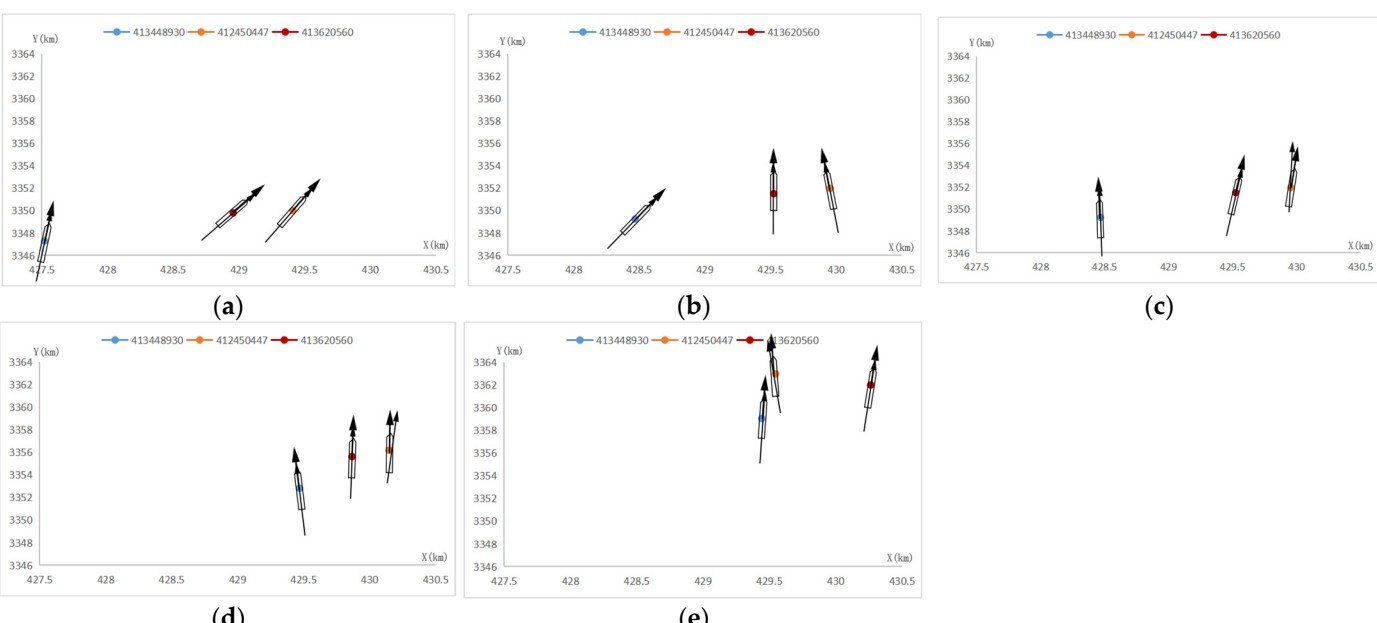

**Figure 11.** Position of three ships: (**a**) the three ships at 19:30; (**b**) the three ships at 19:40; (**c**) the three ships at 19:51; (**d**) the three ships at 20:00; (**e**) the three ships at 20:29.

## 4. Discussion

According to the results of Section 3, the NHI can synthesize the ship parameters and navigation conditions in different encounter situations to achieve the collision risk assessments. Once the ship is triggered by other ships to generate a collision risk field, the NHI will make real-time changes according to the movement status of the ships in the field, and complete the assessment of ship collision risk quickly in combination with the risk level classified in Table 2. With the judgment of collision risk, other ships are monitored to maintain a safe distance and take timely collision avoidance measures. Furthermore, this method can reflect the real-time NHI, whether in the two-ship encounter situation or the multi-ship encounter situation.

In addition, the NHI can be used to avoid collisions. The early real-time collision risk assessment proposed in this paper is the premise of collision avoidance. Because the fields have directions and overlap with each other, we will study the collision avoidance behavior with the intrusion amount of the field as a penalty term in the later stage, and combine the collision avoidance rules at sea and the local harbor chapter as the later research direction. We will generally set an initial safe distance as the latest moving point, and further research will be explained later. We can calculate the NHI value and define the NHI at this time as the critical value. In the process of collision avoidance, depending on when other ships keep the original motion state, the ship can plan a suitable collision avoidance path on the premise that the NHI does not exceed the critical value.

## 5. Conclusions

(1) This paper proposes an algorithm of a navigational hazard index based on field theory, which complements and combines the traditional method of calculating the risk of ship collision and considers the two-ship encounter situation and the multi-ship encounter situation. The problem of domain overlap is transformed into the problem of field overlap. Finally, through the verification of an example, the results show that this method can accurately and stably obtain the NHI in real time and evaluate the risk level around the ship.

(2) The strength of the approach appears in the capability to provide real-time change of the NHI in autonomous ships, so that the ship operating system can judge the current collision risk and take collision avoidance measures quickly. It can be applied

to an on-board anti-collision decision-making system and promotes the automation level of an autonomous ship. It can not only improve the navigation safety of ships at sea, but also provide a reference for the application and development of intelligent navigation technology.

In order to promote the further development of autonomous ships and apply the algorithm to production and practice, future research should focus on improving the feasibility and accuracy of the algorithm, expanding the influencing factors and dimensions of the field (such as considering hydrological conditions), adding time dimensions, and so on.

**Author Contributions:** The authors confirm that the contributions to the paper are as follows: study conception and design: Y.L. and Y.M.; data collection: Y.L. and Y.M.; analysis and interpretation of results: Y.L. and Y.M.; draft manuscript preparation: Y.L. and Y.M. All the authors reviewed the results and approved the final version of the manuscript. All authors have read and agreed to the published version of the manuscript.

**Funding:** This work was supported by the National Natural Science Foundation of China under grant 51509151, and in part by the Shandong Province Key Research and Development Project under grant 2019JZZY020713, the Shanghai Commission of Science and Technology Project under grant 21DZ1201004 and 2300501900, and the Anhui Provincial Department of Transportation Project under grant 2021-KJQD-011.

**Institutional Review Board Statement:** Not applicable.

**Informed Consent Statement:** Not applicable.

**Data Availability Statement:** No new data are created or analyzed in this study. Data sharing is not applicable to this article.

**Conflicts of Interest:** The authors declare no conflict of interest.

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
