# Peer review of "A Field Theory-Based Novel Algorithm for Navigational Hazard Index"

_jmse, doi:10.3390/jmse11010178_

Round 1

Reviewer 1 Report

The manuscript theme is conventional with trivial novelty at the implementation level. Still, a few amendments will enable the manuscript more worthy to suit the wider audience of a journal.

1.       Kindly revise the Title. The Title must be informative representing the methodologies used in the work.

2.       The language of the text especially the abstract is highly adolescent and needs to be carefully rectified with a native language processor instead consult a subject expert in the field.

3.       Incorporate a robust tabular comparison of the proposed analysis on Navigational hazard index based on field theory with other state-of-the-art methodologies/Techniques (on relevant parameters of current interest) recently published in high-impact journals (not older than 2017) in the domain.

4.       Based on the tabular comparison, please mention (point-wise in bullets) the previous drawback/research gaps (as you already mentioned) that motivated you to pursue this study.

5.       Then Highlight your contribution (Point-wise) to addressing the research gap.

6.       Incorporate the complete organization of the article at the end of the introduction section; i.e. How the manuscript has been organized??

7.       The result section is not convincing and still, not exactly supporting the theoretical claim. Incorporate more test cases of interest followed by relevant graphics and numeric analysis in support of test cases. In its current form, the result section is highly poor and doesn’t justify the work properly.

8.       It is better to conclude your work point-wise instead of the running text.

Author Response

Dear  Reviewer:

RE: Manuscript ID: #jmse-2090763 , entitled Navigational hazard index based on field theory.

Thank you for your comments concerning our manuscript. Those comments are all valuable and very helpful for revising and improving our paper, as well as the important guiding significance to our researches.

We have addressed the comments carefully and made revisions accordingly. Revised portions are marked with different colors in the paper.

Specific comments and its responses are as follows:

  1. Kindly revise the Title. The Title must be informative representing the methodologies used in the work.

Thank you for your comments. We have revised the title. The new title is an algorithm of the navigational hazard index based on field theory. And the changes have been annotated in the text.

  1. The language of the text especially the abstract is highly adolescent and needs to be carefully rectified with a native language processor instead consult a subject expert in the field.

Thank you for your comments. We have used the editing service recommended by the magazine to help us modify the language. And the changes have been annotated in the text.

  1. Incorporate a robust tabular comparison of the proposed analysis on Navigational hazard index based on field theory with other state-of-the-art methodologies/Techniques (on relevant parameters of current interest) recently published in high-impact journals (not older than 2017) in the domain.

Thanks for your comments, we have made the robust tabular comparison on line 113. And the changes have been annotated in the text.

  1. Based on the tabular comparison, please mention (point-wise in bullets) the previous drawback/research gaps (as you already mentioned) that motivated you to pursue this study.

Thanks for your comments, the previous drawback/research gaps have been added from line 85 to line 112 .And the changes have been annotated in the text.

  1. Then Highlight your contribution (Point-wise) to addressing the research gap.

Thanks for your comments, we have highlighted our contribution (Point-wise) to addressing the research gap from line 122 to line 135. And the changes have been annotated in the text.

  1. Incorporate the complete organization of the article at the end of the introduction section; i.e. How the manuscript has been organized??

Thanks for your comments, we have added the complete organization of the article from line 136 to 139. And the changes have been annotated in the text.

  1. The result section is not convincing and still, not exactly supporting the theoretical claim. Incorporate more test cases of interest followed by relevant graphics and numeric analysis in support of test cases. In its current form, the result section is highly poor and doesn’t justify the work properly.

Thanks for your comments, We have added some relevant graphics and text descriptions to support the theoretical claim on pages 11 and 12. The ‘discussion’ and ‘conclusion’ have also been modified and improved. And the changes have been annotated in the text.

  1. It is better to conclude your work point-wise instead of the running text.

Thanks for your comments, we have concluded our work point by point. And the changes have been annotated in the text.

Reviewer 2 Report

1. Please revise the first sentence of your abstract as it was poorly written.

2.  Please revise your abstract to ensure it more readable. Include some numbers in your results to show the effectiveness of the proposed work.

3. Please highlight the contribution of the work at the end of the introduction section

4. Typo of crossing in Figure 1. Please check.

5. Make a proper space to separate the equation (13)

6. Please put citation of your Table 1.

7. How the proposed method is able to be translated in the sense it can be easily understood by the person in charge?

8. How do you compare your proposed effectiveness with the existing results.

9. So, the proposed method only able to provide early notification? how about to avoid the collision?

10. The Discussion section is too simple and not significant.

11. Most of the references are old (more than 5 years old). Please include the most recent ones.

Author Response

Dear Reviewer:

RE: Manuscript ID: #jmse-2090763 , entitled Navigational hazard index based on field theory.

Thank you for your comments concerning our manuscript. Those comments are all valuable and very helpful for revising and improving our paper, as well as the important guiding significance to our researches.

We have addressed the comments carefully and made revisions accordingly. Revised portions are marked with different colors in the paper.

Specific comments and its responses are as follows:

  1. Please revise the first sentence of your abstract as it was poorly written.

Thanks for your comments. We have revised the first sentence of our abstract from line 9 to line 11. And the changes have been annotated in the text.

  1. Please revise your abstract to ensure it more readable. Include some numbers in your results to show the effectiveness of the proposed work.

Thank you for your comments. We have used the editing service recommended by the magazine to help us modify the language. And We have added numbers in my results. And the changes have been annotated in the text. 

  1. Please highlight the contribution of the work at the end of the introduction section.

Thank you for your comments. We have highlighted the contribution of the work at the end of the introduction section from line 122 to line 135. And the changes have been annotated in the text.

  1. Typo of crossing in Figure 1. Please check.

Thanks for pointing this out. It has been fixed.

5.Make a proper space to separate the equation (13)
 Thanks for pointing this out. It has been fixed.

  1. Please put citation of your Table 1.

  Thank you for your comments,we have put citation of my Table 1 in the section 3 and 4. And the changes have been annotated in the text.

  1. How the proposed method is able to be translated in the sense it can be easily understood by the person in charge?

  Thank you for your comments. This paper proposes a real-time collision risk assessment method for intelligent ships based on field theory, which can be applied to an on board anti-collision decision-making system and promotes the automation level of an autonomous ship.

  1. How do you compare your proposed effectiveness with the existing results.

  Thank you for your comments. According to the results of Section 3, this method can synthesize the ship parameters and navigation conditions in different encounter situations to achieve the collision risk assessments. And it can be used for real-time dynamic risk assessment in traffic intensive waters, and provides an effective way for intelligent navigation.

  1. So, the proposed method only able to provide early notification? how about to avoid the collision?

  Thank you for your comments. The early real-time collision risk assessment proposed in this paper is the premise of collision avoidance. Because the fields have directions and overlap with each other, we will study the collision avoidance behavior with the intrusion amount of the field as a penalty term in the later stage, and combine the collision avoidance rules at sea and the local harbor chapter as the later research direction.

  1. The Discussion section is too simple and not significant.

Thanks for pointing this out. The ‘Discussion’ is added, which includes comparing the effectiveness of the method with the existing results, providing early notification, and so on. It is shown from line 385 to line 404.

  1. Most of the references are old (more than 5 years old). Please include the most recent ones.

Thanks for pointing this out. The most recent references are added. And the changes have been annotated in the text.

Reviewer 3 Report

The article deals with the issue of maritime navigation safety and solves current problems in this area to achieve minimization of the risk of collision. The authors proposed a methodology for solving these problems, and the results show that the proposed method can overcome the limitations of traditional methods, calculate the index in real time, weaken the influence of the human factor, and provide an effective guarantee of the navigation safety of autonomous ships. Therefore, this method has an important application potential in the development of intelligence navigation.
Water transport appears to be a promising method of transport, especially for large volumes of goods and long transport distances. Therefore, this article is highly relevant and it is necessary to solve the mentioned issue.
In the introduction, the authors analyze the needs for solving this work and at the same time list similar works in this area. The overview of the current status is detailed with links to relevant references.

Comments:
Figure 1 contains "Corssing" grammatical errors. It needs to be fixed.
The title of the 2nd chapter contains an additional point: "2. .Materials and Methods". Likewise, chapter 2.1.
The names of Figure 2, 3, 4 are missing spaces. Also check the other images because almost all of them are without gaps. Dots are always followed by spaces. And there should also be a space after the parenthesis. And somewhere there is a double semicolon. Spaces are also written after semicolons and colons.
Line 132: why are units of quantities written in italics? Do not use italic style for writing units. Also further on in the text, this problem is on line 143, 145.
Figure 3a: What quantities and units are on the X and Y axes?
Figure 5 shows a table in which the units for some quantities are missing.
Figures 6, 8 show characters in the authors' native language. It needs to be translated.
The discussion and conclusion are very brief. A scientific article must have a very detailed conclusion with a critical evaluation of the results achieved. The achieved results, benefits and novelties of this work must be listed here, as well as plans for future research in this area.
References 5, 24, 25, 26, 28 do not have all bibliographic data.
The article contains many grammatical errors. Text revision is needed.

Author Response

Dear Reviewer:

RE: Manuscript ID: #jmse-2090763 , entitled Navigational hazard index based on field theory.

Thank you for your comments concerning our manuscript. Those comments are all valuable and very helpful for revising and improving our paper, as well as the important guiding significance to our researches.

We have addressed the comments carefully and made revisions accordingly. Revised portions are marked with different colors in the paper.

Specific comments and its responses are as follows:

Comments:

Figure 1 contains "Corssing" grammatical errors. It needs to be fixed.

The title of the 2nd chapter contains an additional point: "2. .Materials and Methods". Likewise, chapter 2.1.

The names of Figure 2, 3, 4 are missing spaces. Also check the other images because almost all of them are without gaps. Dots are always followed by spaces. And there should also be a space after the parenthesis. And somewhere there is a double semicolon. Spaces are also written after semicolons and colons.

Line 132: why are units of quantities written in italics? Do not use italic style for writing units. Also further on in the text, this problem is on line 143, 145.

Figure 3a: What quantities and units are on the X and Y axes?

Figure 5 shows a table in which the units for some quantities are missing.

Figures 6, 8 show characters in the authors' native language. It needs to be translated.

The discussion and conclusion are very brief. A scientific article must have a very detailed conclusion with a critical evaluation of the results achieved. The achieved results, benefits and novelties of this work must be listed here, as well as plans for future research in this area.

References 5, 24, 25, 26, 28 do not have all bibliographic data.

The article contains many grammatical errors. Text revision is needed.

Reply:

We are very glad that you recognize our work. Thank you very much.

The figure 1 contains "Corssing" grammatical errors is fixed.

The title of the 2nd chapter is fixed according to the format.

The names of Figure 2, 3, 4 are added spaces.

The italic style for units in the line 132,143,145 is fixed.

The quantities and units on the X and Y axes in the Figure 3a are added .

The units for some quantities in the Figure 5 are added.

The figures 6, 8 are modified by the format.

The ‘Discussion’ is added, which includes comparing the effectiveness of the method with the existing results, providing early notification, and how to avoid collisions. And the ‘Conclusions’ is added, which states the achieved results ,benefits and further research directions. The above contents are from line 384 to line 425.

The reference you mentioned that does not have all bibliographic data is the graduation thesis of Chinese graduate students.We will mark this.

We used the editing service recommended by the magazine to help us modify the language.

Round 2

Reviewer 1 Report

The Authors have professionally addressed the Reviewer's comments. On a round of revision, the manuscript touches the journal standards and may be accepted with a few amendments.

1. Authors may reframe the Title (If they found it worthy for their manuscript) as:

"A Field theory based novel algorithm for navigational hazard index"

2. The language of the manuscript, especially the abstract seems highly twisted and difficult to understand.  Authors are advised to kindly keep the abstract as simple as possible. The abstract is at the forefront of the manuscript and must be simple and informative to suit the wider audience of the journal.

Kindly consult a subject expert; else use a native language processor and carefully proofread before resubmission.

Author Response

Dear Reviewer:

RE: Manuscript ID: #jmse-2090763 , entitled Navigational hazard index based on field theory.

Thank you for your comments concerning our manuscript. Those comments are all valuable and very helpful for revising and improving our paper, as well as the important guiding significance to our researches.

We have addressed the comments carefully and made revisions accordingly. Revised portions are marked with different colors in the paper. 

Specific comments and its responses are as follows:

  1. Authors may reframe the Title (If they found it worthy for their manuscript) as:"A Field theory based novel algorithm for navigational hazard index"

Thank you for your comments. We have revised the title as"A Field theory based novel algorithm for navigational hazard index". And the changes have been annotated in the text.

  1. The language of the manuscript, especially the abstract seems highly twisted and difficult to understand.  Authors are advised to kindly keep the abstract as simple as possible. The abstract is at the forefront of the manuscript and must be simple and informative to suit the wider audience of the journal.

Thank you for your comments. We have checked the language of the manuscript and made changes. And We have made changes to the Abstract. It includes the background, purpose, method, result, and conclusion of the study. And the changes have been annotated in the text.
